# The prevalence of overweight and obesity and the assessment of associated risk factors among school-aged adolescents in Kandahar City, Afghanistan

Mohibullah Mako[1], Najeebullah Rafiqi[2], Azizullah Noori[1], Bilal Ahmad Rahimi [3,4]*

1 Department of Public Health, Faculty of Medicine, Kandahar University, Kandahar, Afghanistan,
2 Department of Surgery, Faculty of Medicine, Kandahar University, Kandahar, Afghanistan, 3 Department of Pediatrics, Faculty of Medicine, Kandahar University, Kandahar, Afghanistan, 4 Department of Public Health, Faculty of Medicine, Afghan International Islamic University, Kabul, Afghanistan

* drbilal77@yahoo.com

## Abstract

### Background

Adolescent obesity is a growing global public health issue, contributing to the early onset of non-communicable diseases like type 2 diabetes and cardiovascular disorders. In South Asia, including Afghanistan, urbanization and lifestyle changes have triggered a nutritional shift marked by unhealthy diets and reduced physical activity. Yet, research on adolescent obesity in Afghanistan, particularly in urban areas like Kandahar, is scarce. Kandahar's rapid development and cultural diversity necessitate an assessment of obesity prevalence and associated risk factors among school-going adolescents to inform health policies.

### Methods

This cross-sectional analytical study was conducted among 384 male adolescents (aged 10–19 years) in Kandahar City between February and July 2023. Height and weight were measured to calculate BMI, which was classified using CDC BMI-for-age percentiles.

### Results

The prevalence of overweight/obesity was 13.3%. Multivariate analysis identified parental obesity (+10.4; p = 0.002), screen time ≥30 minutes/day (+8.8; p = 0.011), and consumption of school canteen food (7.8; p = 0.037) as significant predictors of higher BMI percentiles.

### Conclusion

Targeted interventions involving Family-centered education, promotion of active lifestyles, and regulation of school nutrition are critical to address adolescent obesity in

**Data availability statement:** All relevant data are within the paper and its Supporting Information files.

**Funding:** The author(s) received no specific funding for this work.

**Competing interests:** The authors have declared that no competing interests exist.

this setting. Findings are limited to male adolescents due to cultural constraints and may not be generalizable to females.

## Introduction

Overweight and obesity among children (0–12 years) and adolescents (13–19 years) have become critical global public health issues due to their strong association with chronic non-communicable diseases (NCDs) in adulthood [1]. Numerous studies have linked early-life obesity to serious health conditions, including renal disorders, dyslipidemia, hypertension, cardiovascular diseases, metabolic syndrome, glucose intolerance, and type 2 diabetes mellitus [2–4].

The global burden of childhood and adolescent obesity has escalated markedly in recent decades. In 2008, approximately 170 million individuals under 18 years were classified as overweight or obese, and projections suggest that around 30% of all children will be affected by 2030 [5,6]. In 2021, 93.1 million (95% uncertainty interval 89.6–96.6) individuals aged 5–14 years and 80.6 million (78.2–83.3) aged 15–24 years had obesity [7].

High-income countries are particularly impacted. In the US, over 60% of adults are overweight and more than 30% are obese, while childhood obesity affects over 17% of the pediatric population [8,9]. Similarly, in the UK, 23% of men and 25% of women are obese, and among children aged 2–15, 5.5% of boys and 7.2% of girls are obese, with significantly more categorized as overweight [10]. Across Europe, the overall prevalence of overweight (including obesity) among schoolchildren is estimated at 20.5%, and as well as in Eastern Asia and Western Asia, showing rates of 24.5% and 11.9%, respectively [11]. The greatest increase from 1990 to 2021 in the prevalence of overweight and obesity was seen in Southeast Asia, the Middle East (e.g., United Arab Emirates and Kuwait), East Asia, and Oceania (e.g., Taiwan, Maldives, and China) [7].

Interestingly, low- and middle-income countries are now exhibiting similar upward trends in childhood obesity. These nations face a dual burden, where undernutrition and overnutrition coexist within the same households [12]. Rapid urbanization is changing the food habits, resulting in socioeconomic, demographic, and cultural changes leading to nutritional transition in low-income countries [12,13].

Afghanistan has not been immune to this growing epidemic. A recent study in 2025 involving 2,180 school-aged children in Kandahar reported a combined overweight and obesity prevalence of 11.5% [14]. Despite the rising rates, there remains a paucity of data on adolescent obesity and its contributing risk factors in Afghanistan.

Childhood and adolescence represent a critical window for the prevention and control of obesity and its long-term complications. Generating reliable local data is a solid foundation for designing effective public health interventions and policies.

This study aimed to assess the prevalence of overweight and obesity among adolescents aged 9–17 years in Kandahar, identify associated risk factors, and provide evidence-based recommendations for their prevention and management.

The findings from this study will help inform public health interventions, policies, and school-based programs aimed at reducing the burden of overweight and obesity among Afghan adolescents, with the potential for broader application across the country.

## Materials and methods

### Study design

This cross-sectional analytic study was conducted in twenty schools across Kandahar City, including ten public and ten private schools. Students from grades seven, eight, and nine were selected from these schools. Data were collected through a face-to-face interview using a structured questionnaire. Each student was asked the questions directly, and with the help of the interviewers, their responses were recorded on the questionnaire. The questionnaire was designed to gather information about different types of food intake, physical activity, time spent watching television, and other lifestyle habits.

### Study area

The study was conducted in ten governmental and ten private schools located in various parts of Kandahar City, where students from different economic backgrounds are studying. Kandahar, located in the south of Afghanistan between the Tarnak and Arghandab rivers, lies at an elevation of approximately 1,010 meters above sea level. According to the 2023 estimate, the population of the city is around 543,000, with the majority being Pashtuns. Kandahar is considered one of the most important cities in Afghanistan due to its historical, cultural, and economic significance. It is well known for its rich heritage, agricultural products, and political influence.

### Study population

The participants of the study were male students from both public and private schools in grades seven to nine, aged between 10 and 19 years. Female students could not be included due to cultural and political restrictions in Kandahar at the time of data collection, and thus, the results are representative of male adolescents only. Those who did not agree to take part in the study, those with acute or chronic illnesses, and those whose overweight or obesity was due to medical conditions were excluded from the study.

### Sampling

In the first stage, a complete list of both public and private high and middle schools in Kandahar City was prepared. Then, using random sampling, ten public and ten private schools were selected. To gain access and collect data from students, an official letter was first obtained from the Faculty of Medicine and then from Kandahar University. With this letter, permission was requested from the Department of Education of Kandahar, which issued an approval letter. A copy of this letter was then shared with the administration of each selected school. At the time of data collection, students from grades 7–9 who were present in school were included in the study. Before visiting the schools, the data collectors were trained on how to use the questionnaire and how to accurately measure height and weight. They were shown how to ask each question and how to properly conduct anthropometric measurements.

The number of students selected from each class was determined through systematic random sampling. The selected schools were located in various districts of Kandahar City, and their students represented different social and economic backgrounds.

Before starting the study, an approval letter was also obtained from the Research Committee of Kandahar University. A pre-structured and pre-tested questionnaire was used to collect data.

Height was measured using a stadiometer and weight using a standardized scale, with participants barefoot and in light clothing. Body Mass Index (BMI) was calculated as weight (kg) divided by height squared (m²). According to CDC

BMI-for-age percentiles, adolescents with BMI between the 85th–95th percentile were classified as overweight, and those ≥95th percentile as obese.

BMI-for-age percentiles were calculated using the CDC growth reference charts (ages 2–19 years) [15]. We chose CDC references over WHO growth standards because (i) they cover the full adolescent age range [15], (ii) they are widely used in regional studies (as in [14,16]), which enable better comparability, and (iii) they are easily applicable for epidemiological analysis in this age group [15]. The CDC growth charts are readily integrated into major statistical packages and epidemiological analysis tools, which facilitates reliable and standardized classification of overweight and obesity in our dataset.

## Data collection

A pre-structured and pre-tested questionnaire was used to collect data during the period of February 1–July 31, 2023. This questionnaire was divided into three sections. The first section gathered sociodemographic information about the students, including age, place of residence, type of school, family type, the parents' education and occupation, and economic status. The second section focused on students' dietary habits and lifestyle, such as types and frequency of food intake, time spent watching television, and outdoor play activities. The third section was related to physical measurements, where each student's weight and height were recorded as previously described.

The number of students selected from each class was determined using a systematic sampling method. During data collection, school principals and teachers helped in identifying and selecting the students. The questionnaires were completed in a separate, quiet room within the school to ensure privacy and comfort. If a student had difficulty understanding a question, it was explained to them before recording the answer. Height and weight were measured using a standard scale and measuring tape, as previously mentioned.

## Data analysis

All collected data were first entered into Microsoft Excel and then analyzed using SPSS Version 22. The data were coded before being entered into the computer for analysis. Both descriptive and analytical statistics were used. To explore the relationships between overweight/obesity and associated factors, correlation tests were applied, including univariate, bivariate, and multivariate analyses. In multivariate analysis, we ran a correlation analysis to rule out multicollinearity.

## Ethical considerations

Each participant was informed about the purpose of the study before filling out the questionnaire, and participation was entirely voluntary. Students were assured of the confidentiality of their personal information. All collected data were coded to ensure anonymity. The technical and ethical aspects of the study were approved by the Research Committee of the Faculty of Medicine. Ethical clearance was obtained from the Ethics Committee of Kandahar University (code number KDRU-EC-2022.12) and permission from the Kandahar Province Education Department authorities.

## Results

In this cross-sectional analytical research, we used univariate, bivariate, and multivariate methods.

## Univariate analysis

As in Table 1, A total of 396 male students were surveyed, with no female students in the selected schools due to cultural and political issues; 384 provided complete responses (96.9%) and were included in the analysis. The mean (SD) age, height, and weight were 14.6 (1.5) years, 159.3 (9.5) cm, and 50.3 (13.3) kg, respectively. Exclusions (n = 12) were due to age > 19 years (n = 5), refusal (n = 4), or incomplete data (n = 3). Participants were drawn from all 15 districts of Kandahar City, with the highest representation from District 1 (13.3%) and District 5 (12.2%), and the lowest from District 8 (1.8%).

**Table 1. Socioeconomic, demographic, and educational characteristics of the study participants (n = 384).**

| Variable | Category | Frequency (n) | Percentage (%) |
|---|---|---|---|
| **Gender** | Male | 384 | 100 |
| | Female | 0 | 0 |
| **Type of School** | Public | 232 | 60.4 |
| | Private | 152 | 39.6 |
| **School Grade** | Grade 7 | 135 | 35.2 |
| | Grade 8 | 135 | 35.2 |
| | Grade 9 | 114 | 29.7 |
| **Academic Performance** | High | 146 | 38.0 |
| | Average | 198 | 51.6 |
| | Low | 40 | 10.4 |
| **Father's Education Level** | Illiterate | 133 | 34.6 |
| | No Formal Education | 33 | 8.6 |
| | Primary | 45 | 11.6 |
| | Secondary | 76 | 19.8 |
| | Higher | 97 | 25.3 |
| **Father's Occupation** | Government Employee | 45 | 11.7 |
| | Private Sector Employee | 105 | 27.3 |
| | Business Owner | 131 | 34.1 |
| | Laborer | 44 | 11.5 |
| | Unemployed | 55 | 14.3 |
| | Deceased | 4 | 1.0 |
| **Source of Drinking Water** | Piped Water | 175 | 45.6 |
| | Covered Well | 205 | 53.4 |
| | Uncovered Well | 3 | 0.8 |
| | Tank Water | 1 | 0.3 |
| **Toilet Facility** | Improved Toilet | 329 | 85.7 |
| | Unimproved Toilet | 52 | 13.5 |
| | Open Defecation | 3 | 0.8 |
| **Source of Household Income** | Agriculture | 35 | 9.1 |
| | Livestock | 1 | 0.3 |
| | Monthly Salary | 90 | 23.4 |
| | Private Work | 252 | 65.6 |
| | Aid/Assistance | 6 | 1.6 |
| **Economic Status of Household** | Good | 161 | 41.9 |
| | Moderate | 172 | 44.8 |
| | Poor | 51 | 13.3 |

Most students (60.4%) attended public schools, while the remaining attended private schools, and academic performance was reported as average in 51.6%, high in 38.0%, and low in 10.4%.

Fathers' education levels ranged from illiterate (34.6%) to higher education (25.3%), with private business (34.1%) and private sector employment (27.3%) as the most common occupations. When it comes to Household water sources and toilets, water sources were primarily covered wells (53.4%) or piped water (45.6%); and 85.7% of the students had improved toilets. Household income was mainly from private work (65.6%) or a monthly salary (23.4%), with 44.8% of families reporting moderate economic status, 41.9% good, and 13.3% poor.

Screen-time analysis showed that, out of 384 adolescents, 180 (46.9%) watched television. For total daily screen exposure across all devices, 33.9% reported no screen use, 20.6% reported 1–29 minutes, 10.2% reported 30–59 minutes, 18.0% reported 60–119 minutes, and 17.4% reported ≥2 hours, as shown in Table 2.

For analytical purposes, screen time was categorized using a cutoff of ≥30 minutes/day. This threshold was based on the data distribution in our sample, as a majority of participants reported very low screen exposure. While international guidelines often use ≥2 hours/day as the risk threshold [17], this higher cutoff was not practical for analysis in our dataset due to the small number of adolescents exceeding that level.

Regarding physical activity, 132 students (34.4%) were not engaged in any form of exercise. However, 99 (25.8%) were exercising for 1–29 minutes daily, 60 (15.6%) for 60–119 minutes, and 21 students (5.5%) were engaged in physical activity for more than 2 hours per day, as summarized in Table 2.

According to the findings in Table 3, 136 students (35.4%) had at least one parent who was currently obese. A total of 54 students (14.1%) were taking medications, and 78 (20.3%) had a history of chronic illness.

Based on CDC Percentile classification, BMI value ranged from 1 to 99.99 percentiles, with a mean BMI percentile of 42.26 and a standard deviation of 32.01. Based on CDC criteria, adolescents were divided into four BMI percentile groups, as shown in Table 4. However, for simplification in analysis, these four groups were merged into two main categories: Group

Table 2. Characteristics of leisure and hobby activities of the study participants (n = 384).

| Variable | Category | Frequency (n) | Percentage (%) |
|---|---|---|---|
| **Watching TV** | Yes | 180 | 46.9 |
| | No | 204 | 53.1 |
| **Time Spent Watching TV** | Never | 204 | 53.1 |
| | 1–29 minutes | 55 | 14.3 |
| | 30–59 minutes | 67 | 17.4 |
| | 60–119 minutes | 33 | 8.6 |
| | More than 2 hours | 25 | 6.5 |
| **Time Spent on Screen Devices** | Never | 130 | 33.9 |
| | 1–29 minutes | 79 | 20.6 |
| | 30–59 minutes | 39 | 10.2 |
| | 60–119 minutes | 69 | 18.0 |
| | More than 2 hours | 67 | 17.4 |
| **Time Spent Doing Physical Activity** | Never | 132 | 34.4 |
| | 1–29 minutes | 99 | 25.8 |
| | 30–59 minutes | 72 | 18.8 |
| | 60–119 minutes | 60 | 15.6 |
| | More than 2 hours | 21 | 5.5 |

Table 3. Family history of obesity, other diseases, and taking medication in the study participants (n = 384).

| Variable | Category | Frequency (n) | Percentage (%) |
|---|---|---|---|
| **Parental Obesity** | Yes | 136 | 35.4 |
| | No | 248 | 64.6 |
| **Presence of Other Diseases** | Yes | 78 | 20.3 |
| | No | 306 | 79.7 |
| **Currently Taking Medication** | Yes | 54 | 14.1 |
| | No | 330 | 85.9 |

**Table 4. Nutritional status of the study participants.**

| Category | Definition | Frequency (n) | Percentage (%) |
|---|---|---|---|
| Underweight | weight less than the 3$^{rd}$ percentile | 48 | 12.5 |
| Normal Weight | Weight between the 3$^{rd}$ and 84.9$^{th}$ percentile | 285 | 74.2 |
| Overweight | Weight between the 85$^{th}$ and 94.9$^{th}$ percentile | 23 | 6.0 |
| Obese | Weight at or above the 95$^{th}$ percentile | 28 | 7.3 |
| Total | — | 384 | 100 |

1 included underweight and normal-weight individuals, while Group 2 included overweight and obese individuals. Out of the total 384 adolescents, 51 fell into the overweight/obese group, and 333 were in the normal/underweight group.

As shown in Table 5, 178 students (46.4%) bought food from the school canteen, 37 (9.6%) brought food from home, and 165 (43%) did not eat anything at school. In terms of meal frequency, 366 students (95.3%) ate 2–5 times

**Table 5. Dietary characteristics of the study participants (n = 384).**

| Variable | Category | Number | Percentage (%) |
|---|---|---|---|
| **Source of Food at School** | Home-prepared food | 37 | 9.6 |
| | Purchased from school canteen | 178 | 46.4 |
| | Fruit | 4 | 1.0 |
| | Nothing | 165 | 43.0 |
| **Number of Daily Meals** | Once | 4 | 1.0 |
| | 2–5 times | 366 | 95.3 |
| | More than 5 times | 14 | 3.7 |
| **Snacks at School** | Home-prepared snacks | 37 | 9.6 |
| | Purchased from school canteen | 178 | 46.4 |
| | Fruit | 4 | 1.0 |
| | Nothing | 165 | 43.0 |
| **Frequency of Consuming Market Snacks** | Always | 36 | 9.4 |
| | Most of the time | 33 | 8.6 |
| | Sometimes | 123 | 32.0 |
| | Occasionally | 183 | 47.7 |
| | Never | 9 | 2.3 |
| **Frequency of Consuming Cold Beverages** | Always | 34 | 8.9 |
| | Most of the time | 32 | 8.3 |
| | Sometimes | 92 | 23.9 |
| | Occasionally | 208 | 54.2 |
| | Never | 18 | 4.7 |
| **Frequency of Eating Sweets** | Always | 72 | 18.8 |
| | Most of the time | 65 | 16.9 |
| | Sometimes | 121 | 31.5 |
| | Occasionally | 123 | 32.0 |
| | Never | 3 | 0.8 |
| **Weekly Intake of Fruits and Vegetables** | Never | 10 | 2.6 |
| | Once | 187 | 48.7 |
| | Twice | 115 | 30.0 |
| | Three times | 57 | 14.8 |
| | More than three times | 15 | 3.9 |

a day, 14 (3.6%) ate more than five times a day, and only 4 (1%) had meals once a day. Regarding food sources at school, 37 brought food from home, 178 used the school canteen, 4 only ate fruits, and 165 did not eat anything during school hours.

Concerning junk food consumption, 36 students (9.4%) always, 33 (8.6%) frequently, 123 (32%) sometimes, 183 (47.7%) occasionally, and 9 (2.3%) never consumed such foods. For cold beverages, 34 (8.9%) always consumed, 32 (8.3%) frequently, 92 (24%) sometimes, 208 (54.2%) occasionally, and 18 (4.7%) never consumed them. When it came to sweets, 72 (18.8%) always ate them, 65 (16.9%) frequently, 121 (31.5%) sometimes, 123 (32%) occasionally, and only 3 (0.8%) never consumed sweets. As for fruit and vegetable intake, 10 students (2.6%) reported never eating them throughout the week. However, 187 (48.7%) consumed them once weekly, 115 (29.9%) twice weekly, 57 (14.8%) three times weekly, and 15 (3.9%) more than three times a week.

Since the study participants were initially divided into four groups based on weight status, for easier analysis, these groups were re-categorized into two groups in our study: underweight and normal weight were combined into one group, while overweight and obese were combined into another group. Out of the 384 adolescent boys in our study, 51 fell into the overweight and obese group, while 333 did not fall into that group.

## Bivariate analysis

In the bivariate analysis, we assessed the association of categorical independent variables with the group of adolescents who were either overweight and/or obese or not. From this analysis, we found that three variables showed a statistically significant association with adolescent overweight and obesity: screen time, consumption of food from the school canteen, and parental obesity (current obesity status of mother or father). The associations of other independent variables with adolescent overweight and obesity were not statistically significant Table 6.

**Table 6.  Association between various independent variables and overweight/obesity among adolescents using the Chi-square test (n = 384).**

| Variable | Overweight/Obese (n = 51) | Normal/Underweight (n = 333) | Total (n) | *P*-value |
|---|---|---|---|---|
| **Screen Time** | | | | **0.001** |
| - Less than 30 minutes or none | 17 | 192 | 209 | |
| - 30 minutes or more | 34 | 141 | 175 | |
| **Eating at School** | | | | **0.005** |
| - Does not eat | 18 | 188 | 206 | |
| - Eats at school | 33 | 145 | 178 | |
| **Cold Drink Consumption** | | | | 0.06 ($\chi^2 = 3.379$) |
| - Rarely or never | 24 | 202 | 226 | |
| - Usually or always | 27 | 131 | 158 | |
| **Parental Obesity** | | | | **0.001** |
| - Not obese | 22 | 226 | 248 | |
| - Obese | 29 | 107 | 136 | |
| **Physical Activity** | | | | 0.4 |
| - Less than 30 minutes or none | 28 | 203 | 231 | |
| - 30 minutes or more | 23 | 130 | 153 | |
| **Watching Television** | | | | 0.21 |
| - No | 23 | 181 | 204 | |
| - Yes | 28 | 152 | 180 | |

## Multivariate analysis

Since one of our objectives was to identify risk factors associated with overweight and obesity among school adolescents, we performed a multiple linear regression test for this purpose. In this test, only three variables were found to be statistically significant risk factors for overweight and obesity among adolescents, which are explained below.

When controlling for all other independent variables, adolescents whose mother and/or father currently has overweight or obesity have, on average, a BMI percentile that is 10.38 points higher than those whose parents do not have overweight or obesity. This difference is statistically significant (*p*-value = 0.002).

Similarly, when controlling for all other independent variables, adolescents who use screens for 30 minutes or more have, on average, a BMI percentile that is 8.77 points higher than those who use screens for less than 30 minutes. This difference is statistically significant (*p*-value = 0.011).

Additionally, regarding consumption of food from the school canteen, when controlling for other independent variables, adolescents who eat food from the school canteen have, on average, a BMI percentile that is 7.82 points higher than those who do not. This difference is also statistically significant (*p*-value = 0.037).

It is worth noting that other important variables included in this model, such as consumption of cold beverages, physical exercise, watching television, consumption of fruits and vegetables, type of school, presence of chronic diseases, daily meal frequency, and some other variables, were not found to be statistically significant in the multiple linear regression analysis. Variables that reached statistical significance in bivariate analysis (p < 0.05) were considered for inclusion in the multivariate model. Borderline associations (e.g., cold drink consumption, p = 0.06) and non-significant factors (e.g., physical activity, p = 0.4) were excluded to avoid overfitting. The three types of analysis are shown in Table 7.

## Discussion

This study aimed to determine the prevalence of overweight and obesity and identify related risk factors among school adolescents in Kandahar City, the second largest city of Afghanistan. Our study included 384 male adolescents, and their BMI was calculated according to CDC standards based on age and gender. The results showed that 13.3% of the participants fell into the overweight/obese category, while the rest belonged to the underweight/normal weight group. Our study focused exclusively on male adolescents (10–19 years) and applied multivariate analysis to identify specific lifestyle predictors (screen time, canteen food, parental obesity), providing more targeted insights into this subgroup. These findings cannot be generalized to female students, who may have different prevalence rates.

**Table 7. Table of coefficients, the association between dependent and independent variables with overweight and obesity (n = 384).**

| Variables | Unstandardized | | Standardized | t | *P*-value |
|---|---|---|---|---|---|
| | B | Std. Error | Beta | | |
| (Constant) | 10.8 | 16.7 | — | 0.7 | 0.518 |
| Current parents' obesity | 10.4 | 3.4 | 0.2 | 3.08 | **0.002** |
| Eating from school canteen | 7.8 | 3.7 | 0.1 | 2.09 | **0.037** |
| Screen time | 8.8 | 3.4 | 0.1 | 2.6 | **0.011** |
| Cold drink | −1.9 | 3.3 | −0.03 | −0.6 | 0.564 |
| Exercise | −2.5 | 3.4 | −0.04 | −0.7 | 0.462 |
| TV time | 0.2 | 3.5 | 0.004 | 0.07 | 0.944 |
| Fruit and vegetable | 3.4 | 4.2 | 0.04 | 0.8 | 0.416 |
| Type of school | 5.2 | 3.7 | 0.08 | 1.4 | 0.163 |
| Underlying disease | 4.4 | 4.0 | 0.06 | 1.1 | 0.271 |
| Meal frequency | 4.5 | 7.4 | 0.03 | 0.6 | 0.546 |

Our prevalence (13.3%) is consistent with the 11.5% previously reported in Kandahar [14], and comparable to ranges observed in Pakistan (5–15%) [16] and Iran (5–13%) [18], though lower than rates from East Asia and the Middle East, where some studies report >20% [19–21]. In India, Amidu and colleagues found obesity and overweight rates of 7.5% and 9.8%, respectively [22]. The differences in prevalence rates among these studies might be due to population differences, as well as the demography of overweight/obesity may differ according to cultural, structural, and ecological factors of the population [23].

In our study, three statistically significant factors were associated with overweight and obesity. One important factor was a family history of obesity, specifically parental obesity. A similar study in Kandahar on 2,281 students also found this factor significant [14]. Our neighboring countries have reported comparable findings; in Pakistan, children of obese mothers had an obesity rate of 28%, compared to 8% in children of non-obese mothers. Similarly, children of obese fathers had a 24% obesity rate versus 9% for children of non-obese fathers [16]. In China, parental obesity was also identified as a significant factor [24]. Parental obesity likely reflects both genetic predisposition and shared lifestyle factors, consistent with regional evidence [16,24]. Alongside social, economic, and environmental factors influences variations in obesity rates across populations [25].

Another significant factor associated with overweight and obesity was screen time. A study in Yazd, Iran, involving 510 students aged 12–16 years, found that 70.3% spent more than two hours per day on screens [26]. A Study in Pakistan also confirmed screen time as a statistically significant factor for obesity and overweight in adolescents [16]. With technological advancement, children spend more time on electronic devices, which reduces time for physical activity. Technology also disrupts sleep-wake cycles, leading to poor sleep habits and dietary disturbances [27]. A study on Canadian children showed that access and nighttime use of electronic devices reduced sleep duration, which contributed to weight gain, poor diet quality, and reduced physical activity [28]. In the United States, 35.3% of children were obese or overweight, and 44% spent more than two hours daily on screens. Increased screen time was linked with physical inactivity, with inactive children having twice the risk of overweight/obesity compared to active peers [29].

Eating food from the school canteen was another important factor linked to overweight and obesity. Our study showed that students who consumed food from the school canteen had, on average, a BMI percentile 7.82 points higher than those who did not. A study in Pakistan also identified fast food consumption, which is low in nutritional value, as a major contributor to obesity and overweight in children and adolescents [16]. The increase in fast food shops in Asian countries has led to higher consumption of junk food, this also causing dietary changes [30]. The quality of lunch and snacks in schools and childcare centers has become a topic of concern. Children often eat one or two meals in these settings. Although policies exist promoting healthy foods, beverages, and snacks in schools, their effect on children's eating habits and obesity rates has not been demonstrated yet [31]. This may be because these policies focus on quality rather than quantity, which more strongly influences overweight and obese adolescents [32].

To our knowledge, this is the first study in Afghanistan to use multivariate regression to identify lifestyle risk factors for adolescent obesity. Our findings show that even short-duration screen time (≥30 minutes/day) and school canteen food consumption are significant predictors in Kandahar, highlighting local behavioral patterns that differ from international thresholds. This provides new evidence to guide context-specific interventions.

## Strengths and limitations

Main strength of this study was that we conducted a random sampling proportionate to the population of Kandahar City (384 participants out of 600,000 residents). Also, the data were analyzed using three approaches: univariate, bivariate, and multivariate analyses.

This study has several important limitations. First, only male adolescents were included without female students, due to cultural and political restrictions; therefore, the findings cannot be generalized to female students. Second, lifestyle information (diet, screen time, and physical activity) was self-reported, which may be subject to recall bias and social

desirability bias. Third, the study was conducted exclusively in urban schools in Kandahar City, limiting generalizability to rural Afghan populations where lifestyle and dietary factors may differ. Finally, the cross-sectional design precludes establishing causal relationships between identified risk factors and obesity.

Another limitation relates to the categorization of screen time. Our analysis used a ≥ 30 minutes/day threshold, which is lower than the ≥ 2 hours/day commonly recommended by international guidelines [17]. This was necessary due to the low prevalence of prolonged screen exposure in our sample; thus, our results should be interpreted as reflecting relative risk within this specific population distribution. Although cold drink consumption showed a borderline association in bivariate analysis, it did not remain significant in the adjusted model. Similarly, physical activity did not show independent predictive value when controlling for other factors. These exclusions reflect statistical criteria rather than a dismissal of their potential role, and future larger studies may explore them further.

After data analysis, three variables were found to be significantly associated with overweight and obesity among adolescents. Differences in study populations, sampling techniques, analysis methods, and demographic variations across regions can lead to differences in significant variables among different studies. For instance, Rahimi and colleagues identified male gender, studying in private schools, belonging to wealthy families, and parental obesity as independent significant factors [14]. In the Middle East, studies have linked low physical activity, eating breakfast, low vegetable intake, and high consumption of sugary drinks to obesity [33]. A comparison between urban and rural areas has also been mentioned as a relevant factor [19,34]. In Iran, studies identified homework time (HT), gender, family size, and mode of transportation to school as influential variables [26,35]. Likewise, a study in Pakistan has pointed to socioeconomic status and the intake of high-energy foods mixed with vegetables alongside others as contributing factors [16].

## Conclusion

Among 384 male adolescent school students surveyed in Kandahar City, 13.3% were found to be overweight or obese. The key factors significantly associated with overweight and obesity among male adolescents included parental obesity, screen time, and consumption of food from school canteens.

## Recommendations

Based on the findings of this study, several targeted actions are recommended. First, since consumption of food from school canteens was significantly associated with higher BMI percentiles, school authorities and policymakers should regulate canteen menus to ensure the availability of healthier food options and limit the sale of high-calorie, low-nutrient items. Second, as increased screen time was an independent predictor of overweight and obesity, schools should integrate structured physical activity programs into the daily schedule to reduce sedentary behavior and promote active lifestyles. Third, because parental obesity emerged as a major risk factor, family-centered interventions such as parental nutrition education and community-based health promotion are essential to address intergenerational patterns of obesity. Finally, coordinated awareness campaigns at the national level should focus on these identified risk factors to design culturally appropriate and evidence-based strategies for Afghan adolescents.

## Supporting information

**S1 File.  SPSS file of the data.**
(SAV)

## Acknowledgments

We extend our heartfelt appreciation to the administration of Kandahar University, the Directorate of Education in Kandahar, and the honorable school authorities who provided their full support during the implementation of this study. We are also thankful to the data collectors, study participants, and their respected parents for their cooperation and contributions.

## Author contributions

**Conceptualization:** Mohibullah Mako, Najeebullah Rafiqi, Azizullah Noori, Bilal Ahmad Rahimi.

**Data curation:** Mohibullah Mako, Azizullah Noori, Bilal Ahmad Rahimi.

**Formal analysis:** Najeebullah Rafiqi, Bilal Ahmad Rahimi.

**Investigation:** Mohibullah Mako, Azizullah Noori.

**Methodology:** Najeebullah Rafiqi, Bilal Ahmad Rahimi.

**Project administration:** Mohibullah Mako, Bilal Ahmad Rahimi.

**Resources:** Mohibullah Mako, Najeebullah Rafiqi, Bilal Ahmad Rahimi.

**Software:** Najeebullah Rafiqi, Bilal Ahmad Rahimi.

**Supervision:** Mohibullah Mako, Azizullah Noori, Bilal Ahmad Rahimi.

**Validation:** Mohibullah Mako.

**Visualization:** Mohibullah Mako, Bilal Ahmad Rahimi.

**Writing – original draft:** Mohibullah Mako, Najeebullah Rafiqi, Bilal Ahmad Rahimi.

**Writing – review & editing:** Mohibullah Mako, Najeebullah Rafiqi, Azizullah Noori, Bilal Ahmad Rahimi.

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
