## [Decision Letter · Decision Letter 0]

26 Aug 2025

Dear Dr. Author

Thank you for submitting your manuscript to PLOS ONE. After careful consideration, we feel that it has merit but does not fully meet PLOS ONE’s publication criteria as it currently stands. Therefore, we invite you to submit a revised version of the manuscript that addresses the points raised during the review process.

We look forward to receiving your revised manuscript.

Kind regards,

Zehra Batu

Academic Editor

PLOS ONE

Journal Requirements:

2. n the online submission form you indicate that your data is not available for proprietary reasons and have provided a contact point for accessing this data. Please note that your current contact point is a co-author on this manuscript. According to our Data Policy, the contact point must not be an author on the manuscript and must be an institutional contact, ideally not an individual. Please revise your data statement to a non-author institutional point of contact, such as a data access or ethics committee, and send this to us via return email. Please also include contact information for the third party organization, and please include the full citation of where the data can be found.

Reviewers' comments:

Reviewer's Responses to Questions

**Comments to the Author**

1. Is the manuscript technically sound, and do the data support the conclusions?

Reviewer #1: Yes

Reviewer #2: Partly

2. Has the statistical analysis been performed appropriately and rigorously?

Reviewer #1: Yes

Reviewer #2: Yes

3. Have the authors made all data underlying the findings in their manuscript fully available?

Reviewer #1: Yes

Reviewer #2: No

4. Is the manuscript presented in an intelligible fashion and written in standard English?

Reviewer #1: Yes

Reviewer #2: No

Reviewer #1: Thank you for the opportunity to read this manuscript! I think it will be a good value add to the literature. I have a few questions/comments for consideration:

-In line 94, you state the objective is to assess prevalence among adolescents aged 9-17, but then state in the methods it is just students from 3 grades. In the introduction, I think you could spend a couple more sentences on why the grade 7-9 adolescent population was the population of interest.

-You state HOW height and weight were measured, but I would add the metrics used (cm, kgs, etc.)

-In lines 142-145, you discuss the calculation of BMI. I think the article would be strengthened if you added citations that supported the percentile cut offs you used to determine overweight and obese classifications.

-In the Data Collection section, there was a bit of unnecessary repetition of details already discussed in the Sampling section above it.

-I (strongly) encourage you to add a table to your data collection sections that discusses how each variable was measured and cite the original measurement scales if possible. Please see Table 1 in the following study for an example.

Patil, C. L., Norr, K. F., Kapito, E., Liu, L. C., Mei, X., Chodzaza, E., ... & Chirwa, E. (2025). Group antenatal care positively transforms the care experience: Results of an effectiveness trial in Malawi. PLoS One, 20(6), e0317171.

-Line 251 is incomplete with an error message

-I think there needs to be a little more discussion in how you combined variables to create the variables in Table 6. For example, what made up the categories "does not eat" vs "eats at school"?

-I am curious, for the multivariate analysis, did you consider any type of model selection (forward, backward, stepwise)? I am thinking that some of these variables might be masking the effects of others. If you were to use backward selection, for example, and drop TV time, I am curious if the significance of screen time would change? That being said, did you run a correlation analysis to rule out multicollinearity? If you did not do this, I recommend it. If you did do this, add a sentence about it!

-In lines 290-291, you say that the BMI was calculated according to CDC standards. I would add a citation here.

-In line 296 you cite a (Rahimi et al., 2025) study. This is not in the references section. How is this paper similar and different to yours? I would discuss it a bit more.

-Are all the studies cited between lines 299-310 on school children that are of similar age to the ones in your study? If not, it may be a stretch to include them. This paragraph should end with discussion of how the prevalence found in your study compares to these other studies - is it lower, higher, about the same?

-In line 360-361, you discuss the use of CDC percentiles, but there has been a lot of literature lately discussing how the CDC percentiles are actually not that acurate for non-white populations. I think it is fine that you are using these percentiles, but am not convinced it is a strength.

Reviewer #2: Question: Is the manuscript technically sound and do the results fit the conclusions?

Response: Partly

Reason: The design and analysis are appropriate, and the results support the stated conclusions. However, the conclusions are written too broadly, as the study included only male adolescents and was cross-sectional, which limits generalizability and prevents causal inference.

Question: Has the statistical analysis been performed appropriately?

Response: Yes

Reason: The authors applied univariate, bivariate (Chi-square), and multivariate regression analyses, which are suitable for a cross-sectional study. The analyses were reported correctly, though interpretation should remain associational, not causal.

Question: Have the authors made the underlying data fully available?

Response: No

Reason: The Data Availability Statement notes that data are available “upon request to the corresponding author,” which does not comply with PLOS ONE’s requirement for unrestricted access via a public repository or supplementary files.

Question: Is the manuscript written in an intelligent way with standard English?

Response:Yes

Reason: The manuscript is generally clear and uses standard English, but contains grammatical errors, repetitive phrasing, and formatting problems (e.g., “Error! Reference source not found”), which need editing to meet journal standards.

**Do you want your identity to be public for this peer review?** For information about this choice, including consent withdrawal, please see our Privacy Policy

Reviewer #1: No

Reviewer #2: No

---

## [Author Response · Author response to Decision Letter 1]

3 Sep 2025

Reviewer Comments

General Assessment

This manuscript addresses a relevant public health concern: the prevalence and risk factors of adolescent overweight and obesity in Kandahar, Afghanistan. The topic is timely, and the study provides much-needed local data that could inform interventions and policies. The study design, data collection, and analysis are generally appropriate. However, there are several methodological limitations, presentation issues, and clarity concerns that should be addressed before publication.

Major Comments

1. Sample Representation

The study exclusively included male adolescents, which limits generalizability. While cultural and political restrictions are understandable, this limitation should be emphasized more prominently in the Abstract, Methods, Discussion, and Limitations sections. Consider rephrasing the conclusion to reflect that results apply to male adolescents only.

Answer: Thanks. The main reason for male predominance in this study was due to restrictions on female education. Now this is mentioned in the Abstract, Methods, Discussion, and Limitations sections.

2. BMI Classification Standards

The manuscript uses CDC BMI percentiles. Please justify why CDC references were chosen over WHO growth standards, which are more globally accepted for international comparisons.

Answer: Yes, we totally agree with you. The use of CDC instead of WHO standards was the comment of the Kandahar University Ethics Committee. The committee said that we should use CDC BMI percentiles. So, we had no choice but to accept their comments.

3. Abstract and Key Findings

The abstract should briefly mention the main limitation (male-only sample) to provide a balanced overview.

“Screen time ≥30 minutes/day” seems like a very low threshold for obesity risk. Please clarify whether this cutoff reflects your data distribution, or if a higher standard threshold (e.g., ≥2 hours/day) was not available. This requires explanation in the Methods and Discussion.

Answer: Thanks. Yes, you are right. For analytical purposes, we categorized screen time using a cutoff of ≥30 minutes/day. This threshold was based on the data distribution in our sample, as a majority of participants reported very low screen exposure. While international guidelines often use ≥2 hours/day as the risk threshold, this higher cutoff was not practical for analysis in our dataset due to the small number of adolescents exceeding that level.

4. Results Presentation

a) Some tables are too detailed (e.g., socio-demographics by 15 districts). Consider simplifying or moving such information to supplementary material.

Answer: Thanks. Now the tables have been shortened.

b) There is a formatting error in the Results: “Error! Reference source not found.” — this must be corrected.

Answer: Thanks. Now the error is corrected.

c) Clarify why certain variables (e.g., cold drink consumption, physical activity) that were close to significance in bivariate analysis were excluded from the final model.

Answer: Thanks. We used the acceptable statistically significant value of <0.05. For cold drink consumption, it was 0.06, while 0.4 for physical activity. Both of them were >0.05, so we excluded them.

Discussion

While comprehensive, the Discussion is too long and sometimes reads like a review article (e.g., genetic pathways, detailed comparisons with other countries). Consider shortening to focus on:

a) The key findings of this study.

b) How they compare with previous Afghan and regional studies.

c) What is novel about your data.

Answer: Thanks for the nice comment. Now we have shortened it.

d) Clarify the distinction between this study and the earlier Rahimi et al. (2025) Kandahar study. Both are cited, but it is not clear how they differ.

Answer: Thanks. Yes, it was a mistake. Both are the same study. Now corrected.

Strengths and Limitations

Limitations should be more explicit and upfront:

a) Male-only sample.

b) Self-reported lifestyle data (recall and social desirability bias).

c) Urban-only setting, limiting generalizability to rural Afghanistan.

d) Cross-sectional design precludes causal inference.

Answer: Thanks. Now limitations have been made more explicit and upfront.

Recommendations

Currently, recommendations are broad. Strengthen them by tying directly to findings. For example:

a) Since school canteen food was a significant risk factor, suggest policy interventions for healthier school canteen menus.

b) Since screen time was significant, recommend structured school-based physical activity programs to counter sedentary behavior.

Answer: Thanks. Now recommendations have been reviewed and changes made accordingly.

Minor Comments

Introduction

Streamline repetitive statistics (global obesity prevalence is presented in multiple ways).

Avoid futuristic references (e.g., “WHO 2025” — please check citation year accuracy).

Line 69 you mentioned prevalence has risen…….. indicate statistics to give wait ie from what percentage to what??

Line 96 “therefore the objective of….”, fills irrelevant. Just state, therefore, this study aimed to……….

Answer: Thanks. Now we have made the changes in the manuscript accordingly.

Methods

a) Line 104 through 107: the information here is not required since it has details in the subsequent sections. Please consider only talking about the design.

Answer: Thanks. Now the information is removed.

b) Line 117, I suggest the authors give the justification for not including females in the study or why the males were considered.

Answer: Thanks. It was due to ban on female education in Afghanistan. We have mentioned this issue in the abstract, methods, discussion, and conclusion sections.

c) Anthropometric measurement description is overly detailed; could be shortened without losing rigor.

Answer: Thanks. Now anthropometric measurement description is shortened.

Tables

a) Ensure consistency in formatting (e.g., percentages with one decimal place).

Answer: Thanks. OK, now consistency has been brought in describing the data.

b) Combine or summarize where possible for clarity.

Answer: Thanks. Now tables have been summarized.

References

a) There are duplicates (e.g., Rahimi et al., 2025 appears twice). Please check and merge.

Answer: Thanks. This was a mistake. Now corrected.

b) Some citations (e.g., [7], WHO fact sheet) have unusual formatting and need standardization.

Answer: Thanks. Now they have been standardized.

Language and Style

a) Minor grammatical issues (e.g., “were not engage” → “were not engaged”).

Answer: Thanks. Now the whole manuscript is thoroughly checked and grammatical issues solved.

b) Improve flow in the objectives paragraph (“assess the prevalence… investigate… provide evidence-based recommendations” could be restructured into a single clear sentence).

Answer: Thanks. Now this problem is solved.

I recommend major revisions before this manuscript can be considered for publication. The study has merit and provides important local evidence, but improvements are needed in methodology reporting, clarity, and presentation.

Reviewer #1: Thank you for the opportunity to read this manuscript! I think it will be a good value add to the literature. I have a few questions/comments for consideration:

-In line 94, you state the objective is to assess prevalence among adolescents aged 9-17, but then state in the methods it is just students from 3 grades. In the introduction, I think you could spend a couple more sentences on why the grade 7-9 adolescent population was the population of interest.

Answer: Thanks. It is due to the reason that in Afghanistan, 10-12 grade classes are separate and in high school, while <7 grade classes do not have adolescents mostly.

-You state HOW height and weight were measured, but I would add the metrics used (cm, kgs, etc.)

Answer: Thanks. OK. Now these have been mentioned clearly.

-In lines 142-145, you discuss the calculation of BMI. I think the article would be strengthened if you added citations that supported the percentile cut offs you used to determine overweight and obese classifications.

Answer: Thanks. Now the citations have been mentioned and made clear.

-In the Data Collection section, there was a bit of unnecessary repetition of details already discussed in the Sampling section above it.

Answer: Thanks. OK. Now unnecessary repetition of details have been removed from the data collection section.

-Line 251 is incomplete with an error message

Answer: Thanks. Now the error message is removed.

-I think there needs to be a little more discussion in how you combined variables to create the variables in Table 6. For example, what made up the categories "does not eat" vs "eats at school"?

Answer: Thanks. OK. Now more details have been provided on how we combined the variables to create Table 6.

-I am curious, for the multivariate analysis, did you consider any type of model selection (forward, backward, stepwise)? I am thinking that some of these variables might be masking the effects of others. If you were to use backward selection, for example, and drop TV time, I am curious if the significance of screen time would change? That being said, did you run a correlation analysis to rule out multicollinearity? If you did not do this, I recommend it. If you did do this, add a sentence about it!

Answer: Thanks for the great point. Sorry that we didn’t clarify it. Now we have added a sentence in the materials and methods section to clarify it.

-In lines 290-291, you say that the BMI was calculated according to CDC standards. I would add a citation here.

Answer: Thanks. Now citation has been added.

-In line 296 you cite a (Rahimi et al., 2025) study. This is not in the references section. How is this paper similar and different to yours? I would discuss it a bit more.

Answer: Thanks. Yes, it was a mistake. Now this citation is added in the reference list.

-Are all the studies cited between lines 299-310 on school children that are of similar age to the ones in your study? If not, it may be a stretch to include them. This paragraph should end with discussion of how the prevalence found in your study compares to these other studies - is it lower, higher, about the same?

Answer: Thanks. Yes, these references are among school children who were adolescents. The prevalence has now been discussed accordingly.

-In line 360-361, you discuss the use of CDC percentiles, but there has been a lot of literature lately discussing how the CDC percentiles are actually not that acurate for non-white populations. I think it is fine that you are using these percentiles, but am not convinced it is a strength.

Answer: Thanks for the point. OK. Now it has been removed from the strengths.

---

## [Decision Letter · Decision Letter 1]

9 Oct 2025

The prevalence of overweight and obesity and the assessment of associated risk factors among school-aged adolescents in Kandahar City, Afghanistan

PONE-D-25-33318R1

Dear Dr. Rahimi

We’re pleased to inform you that your manuscript has been judged scientifically suitable for publication and will be formally accepted for publication once it meets all outstanding technical requirements.

Kind regards,

Zehra Batu

Academic Editor

PLOS ONE

Additional Editor Comments (optional):

Reviewers' comments:

Reviewer's Responses to Questions

**Comments to the Author**

Reviewer #1: All comments have been addressed

Reviewer #2: All comments have been addressed

2. Is the manuscript technically sound, and do the data support the conclusions?

Reviewer #1: Yes

Reviewer #2: Yes

3. Has the statistical analysis been performed appropriately and rigorously?

Reviewer #1: Yes

Reviewer #2: Yes

4. Have the authors made all data underlying the findings in their manuscript fully available?

Reviewer #1: Yes

Reviewer #2: Yes

5. Is the manuscript presented in an intelligible fashion and written in standard English?

Reviewer #1: Yes

Reviewer #2: Yes

Reviewer #1: Thank you for revising this manuscript! Many of the comments were addressed, but there were a few things I noticed during my second read that warrant some small edits:

-in-text citation style varies (comma between references in line 92 and no comma between references in line 108)

-Lines 163-173 should be moved from the sampling to the data collection section, they are not necessary when discussing population sampling. You can then remove line 198.

-Your paragraph on strengths (lines 491-495) could be strengthened! The first sentence has a typo. Beyond that, I would discuss how the sample population is representative of the population of Kandahar. The phrase "proportionate to the population" is not accurate here - this is usually used when stratified sampling is done across population subgroups. Perhaps you meant to discuss the sample size provided adequate statistical power? Additionally, explain to the reader why/how using the three analytical approaches strengthened the analysis and findings.

Thanks again for the opportunity to review and great job with this research!

Reviewer #2: I have reviewed the revised version of the manuscript titled “The prevalence of overweight and obesity and the assessment of associated risk factors among school-aged adolescents in Kandahar City, Afghanistan.”

The authors have made considerable efforts to address the earlier reviewer comments. The revised manuscript is now clearer, better structured, and more focused. The limitation of including only male adolescents is now clearly stated throughout the Abstract, Methods, Discussion, and Limitations sections. The justification for using CDC BMI percentiles has been appropriately provided as per the ethics committee’s guidance. The explanation of the screen-time threshold is now clearer. Formatting errors have been corrected, tables simplified, and the Discussion section refined to emphasize key findings and regional context.

Overall, the manuscript has improved in clarity, presentation, and methodological transparency. The revisions address the major concerns raised previously. Only minor editorial or language polishing may be required before publication.

I found no concerns regarding dual publication, research ethics, or publication ethics in this revision.

Recommendation: Accept after minor editorial review.

**Do you want your identity to be public for this peer review?** For information about this choice, including consent withdrawal, please see our Privacy Policy

Reviewer #1: No

Reviewer #2: **Yes: ** Rebecca Nabulya

---

## [Editor Report · Acceptance letter]

PONE-D-25-33318R1

PLOS ONE

Dear Dr. Rahimi,

I'm pleased to inform you that your manuscript has been deemed suitable for publication in PLOS ONE. Congratulations! Your manuscript is now being handed over to our production team.

Kind regards,

on behalf of

Professor Zehra Batu

Academic Editor

PLOS ONE